# Explainable deep learning for disease activity prediction in chronic inflammatory joint diseases

**Cécile Trottet**[1], **Ahmed Allam**[1], **Aron N. Horvath**[1], **Axel Finckh**[2], **Thomas Hügle**[3], **Sabine Adler**[4,5], **Diego Kyburz**[6], **Raphael Micheroli**[7], **Michael Krauthammer**[1,8☯*], **Caroline Ospelt**[7☯]

**1** Department of Quantitative Biomedicine, University of Zurich, Zurich, Switzerland, **2** Division of Rheumatology, Department of Medicine, Faculty of Medicine, Geneva University Hospitals, Geneva, Switzerland, **3** Department of Rheumatology, Lausanne University Hospital, Lausanne, Switzerland, **4** Department of Rheumatology and Immunology, Kantonsspital Aarau, Aarau, Switzerland, **5** Department of Rheumatology and Immunology, Inselspital - University Hospital Bern, Bern, Switzerland, **6** Department of Rheumatology, University Hospital Basel, Basel, Switzerland, **7** Center of Experimental Rheumatology, Department of Rheumatology, University Hospital Zurich, University of Zurich, Zurich, Switzerland, **8** Biomedical Informatics DFL, University Hospital Zurich, University of Zurich, Zurich, Switzerland

☯ These authors contributed equally to this work.
* michael.krauthammer@uzh.ch

**Data Availability Statement:** Data are owned by a third party, the Swiss Clinical Quality Management in Rheumatic Diseases (SCQM) foundation and

## Abstract

Analysing complex diseases such as chronic inflammatory joint diseases (CIJDs), where many factors influence the disease evolution over time, is a challenging task. CIJDs are rheumatic diseases that cause the immune system to attack healthy organs, mainly the joints. Different environmental, genetic and demographic factors affect disease development and progression. The Swiss Clinical Quality Management in Rheumatic Diseases (SCQM) Foundation maintains a national database of CIJDs documenting the disease management over time for 19'267 patients. We propose the Disease Activity Score Network (DAS-Net), an explainable multi-task learning model trained on patients' data with different arthritis subtypes, transforming longitudinal patient journeys into comparable representations and predicting multiple disease activity scores. First, we built a modular model composed of feed-forward neural networks, long short-term memory networks and attention layers to process the heterogeneous patient histories and predict future disease activity. Second, we investigated the utility of the model's computed patient representations (latent embeddings) to identify patients with similar disease progression. Third, we enhanced the explainability of our model by analysing the impact of different patient characteristics on disease progression and contrasted our model outcomes with medical expert knowledge. To this end, we explored multiple feature attribution methods including SHAP, attention attribution and feature weighting using case-based similarity. Our model outperforms temporal and non-temporal neural network, tree-based, and naive static baselines in predicting future disease activity scores. To identify similar patients, a *k*-nearest neighbours regression algorithm applied to the model's computed latent representations outperforms baseline strategies that use raw input features representation.

may be obtained after approval and permission from SCQM. The code developed for the analysis is available on the following GitHub repository https://github.com/uzh-dqbm-cmi/scqm.

**Funding:** MK was awarded a grant from the Swiss National Science Foundation (project 201184) for this work (https://www.snf.ch/en). The funders had no role in study, analysis, decision to publish, or preparation of the manuscript. The SCQM Foundation is supported by pharmaceutical industries and donors. A list of financial supporters can be found on www.scqm.ch/en/partners/. SCQM supporting partners had no role in the study design or in the analysis and interpretation of the data, the writing of the manuscript or the decision to submit the manuscript for publication.

**Competing interests:** The authors have declared that no competing interests exist.

## Author summary

Chronic inflammatory joint diseases affect about 200′000 patients in Switzerland alone. These conditions lead to immune system dysfunction resulting in inflammation that targets the joint tissues. Understanding which aspects of patients' characteristics and disease management history are predictive of future disease activity is crucial to improving patients' quality of life. A significant obstacle to the widespread adoption of deep learning (DL) methods in healthcare is the challenge of understanding their "black-box" nature (i.e. the underlying decision process for outcome generation). Therefore, the development of "explainable" deep learning methods has become an active area of research. These approaches aim to provide insights into the inner workings of deep learning models, enabling physicians to understand and assess the output of DL models more effectively. We propose DAS-Net: an explainable deep learning model that finds similar patients and predicts future disease activity based on past patient history. In our analysis, we contrast different explainability approaches highlighting which aspects of the patient history impact model predictions the most. Furthermore, we show how computed patient similarities allow us to rank different patient characteristics in terms of influence on disease progression and discuss how case-based explanations can enhance the transparency of deep learning solutions.

## 1 Introduction

Chronic inflammatory joint diseases (CIJDs) cause the immune system to attack healthy organs, particularly the joints [1]. In addition to causing pain, the inflammation can lead to synovitis, bone erosions, muscle and ligament damage. To this day, there exists no cure and the treatments primarily help attenuate the patients' symptoms and improve their quality of life. Finding ways to minimise the disease activity is crucial to alleviate the disease burden on patients' everyday life.

Digitalising patient healthcare data has led to a massive increase in available electronic health records (EHRs), opening up the opportunity to mine these records and employ machine learning (ML) approaches to discover novel evidence about real-world treatment efficacy and patient outcomes [2]. Due to the complex patient-specific disease progression patterns, CIJDs patient registries are very heterogeneous in the collected measurements and temporally sparse, presenting a challenge for ML models to learn from the data. In this work, we use the database of the Swiss Clinical Quality Management in Rheumatic Diseases (SCQM) Foundation [3]. It is a national longitudinal database of CIJDs documenting the disease management over time for 19'267 patients with different forms of arthritis.

We propose the Disease Activity Score Network (DAS-Net), an explainable multi-task neural network model to transform heterogeneous longitudinal patient journeys from the SCQM registry into comparable representations and predict future disease activity scores (DAS). DAS-Net evaluates the importance of the different aspects of individual management history (events) to predict future disease activity scores (i.e. multi-task forecasting). To this end, we trained our model on patients who had available DAS28-BSR (hereafter DAS28) [4] or ASDAS-CRP (hereafter ASDAS) [5] scores, without limiting our analysis to a specific arthritis subtype, but rather including all the patients for which either of these scores was available. The model is composed of multilayer perceptrons, long short-term memory networks [6], and augmented with attention mechanism [7] to process heterogeneous patient histories. The

attention mechanism highlights parts of the patients' histories that are most likely contributing to the outcome prediction, providing further insights into the model's decision-making process.

Compared to physicians who use their experience to assess possible similarities among patients [8], we use our model to retrieve patients with similar disease progression by mapping the patients' raw entangled data into a latent space with higher separability [9]. We empirically assessed DAS-Net's ability to cluster patients with similar disease progressions.

Lastly, we explored multiple explainability approaches in our analysis, in particular through the (a) SHAP (SHapley Additive exPlanations) [10] value computation on the baseline models' input features to gain post-hoc insights into the contribution of each feature (b) two-layered attention mechanism in the model architecture assigning weights to the different events of the patient histories and highlighting their significance for the model's predictions, and (c) case-based importance weighting of the features for patient similarity assessment. We offer visual insights to illustrate how the model evaluates the similarity between some example patients and highlight the most influential features. To expand on these case-based explanations, we developed aggregate metrics to rank the input features' importance for similarity assessment.

By contrasting the results of these various approaches, we believe that we make a significant step towards enhancing the transparency of the model's output.

## 1.1 Related work

Temporal deep learning models such as recurrent neural networks and transformers are commonly used in deep learning to analyse longitudinal patient data [11]. However, there is limited research on employing these temporal modeling approaches to predict disease progression in CIJDs. Most existing DL studies using CIJD databases focus on classifying the diagnoses rather than predicting how the disease progresses [2]. In studies that do predict disease progression, the continuous DAS values are usually simplified and thresholded into a binary classification task such as remission/no remission or response/no response, rather than predicted through regression [12]. For instance, Norgeot et al. [13] implemented RNNs to predict disease activity (remission/no remission) at the next rheumatology visit for rheumatoid arthritis patients. Their model significantly outperformed a static baseline, indicating the effectiveness of employing temporal models for modeling disease activity in CIJDs.

Furthermore, the majority of the existing studies are limited to patients with rheumatoid arthritis. However, in [14], both rheumatoid arthritis (RA) and axial spondyloarthritis (axSpa) patients were included and various non-temporal ML models (such as random forest, logistic regression and vanilla neural networks) were used to predict response/no response to different treatments. Their feature importance analysis revealed that different patient-reported outcome measures were the most significant predictors. This result supports our findings that past measures of disease activity are highly predictive of disease progression.

Our model architecture builds on the work proposed in [15] and further extends it (a) to support patients with different CIJD subtypes (not only RA) and (b) adding attention layers to measure the importance of different patient characteristics and management strategies for the model predictions. To the best of our knowledge, this is the only study emphasising patient similarity and explainability in modeling temporal disease progression in CIJDs.

## 2 Materials and methods

### 2.1 Dataset

**2.1.1 Description.** The SCQM Foundation maintains a national database of inflammatory rheumatic diseases since 1997. The database documents the disease management over time for

19′267 patients through clinical measurements during the visits, demographics, prescribed medications and patient-reported outcome measures (database snapshot from 01.04.2022). Patients are diagnosed either with rheumatoid arthritis (RA), axial spondyloarthritis (axSpA), psoriatic arthritis (PsA) or undifferentiated arthritis (UA). Appendix S1 Fig shows the distribution of the number of medical visits per patient in the database.

**2.1.2 Ethics.** Pseudonymised data, without access to the code key, was provided by the Swiss Clinical Quality Management in Rheumatic Diseases registry to the researchers. Therefore, the ethics commission of the Canton of Zurich (KEK-ZH) waived the need for a full ethics authorization (Declaration of non-responsibility from the KEK-ZH). The SCQM Foundation operates a national register for inflammatory rheumatic diseases in close cooperation with the Swiss Society for Rheumatology SGR. The SCQM Foundation is obliged that all data are subject to federal and/or cantonal data protection regulations. Prior enrolment at SCQM, signed informed consent was provided by the patients, in accordance with the Declaration of Helsinki. Additionally, withdrawal of participation is possible at any time.

**2.1.3 Preprocessing.** The SCQM database documents the management and disease evolution of the patients spanning several types of records and sources. We kept four distinct sources of information:

1. **Demographics (Dem)**: Non-temporal patient features such as date of birth or gender.

2. **Clinical measures (CM)**: Clinical measurements collected during a visit, such as DAS or weight.

3. **Medications (Med.)**: Features related to a prescribed medication and its duration (i.e. start or stop).

4. **Patient-reported outcome measure (PROM)**: Patient self-reported disease activity scores (such as RADAI score [16]).

While the demographics are static and only collected once, the clinical measures, medications and PROM are low-frequency time series. We refer to these as "time-related events".

As preprocessing steps, we discarded patients with less than three CMs with distinct measurements of ASDAS or DAS28, or no medication information. We also discarded records with missing dates in the time-related data, and the clinical measures without either DAS28 or ASDAS. We selected the features used in [15], and additional ones based on availability and clinical relevance. We included the 90% most prescribed medications. After preprocessing, 10′589 patients (with a total of 79′872 clinical measures) and 31 features remained. The list of features is shown in appendix S1 Table and Fig 1 shows the distribution of the two DAS we used as predictive targets (i.e. outcomes). Summary statistics of the features are available in the tables of appendices S2, S3, S4, S5, S6, S7, S8 and S9 Tables. Moreover S2 and S3 Figs show the distribution of the different input features and targets, stratified on the number of medical visits.

## 2.2 Model

**2.2.1 Motivation.** Our dataset, like many EHR datasets, is irregular in both the temporal aspect (patients do not have the same number of medical visits), and in the number of recorded features (patients have varying numbers of recorded measurements and missing attributes).

Using non-temporal machine learning approaches (i.e. models that ignore patients' full history) would limit the modeling of the data by restricting the input features to the subset shared by most data points or by discarding and imputing features to homogenise the data. This

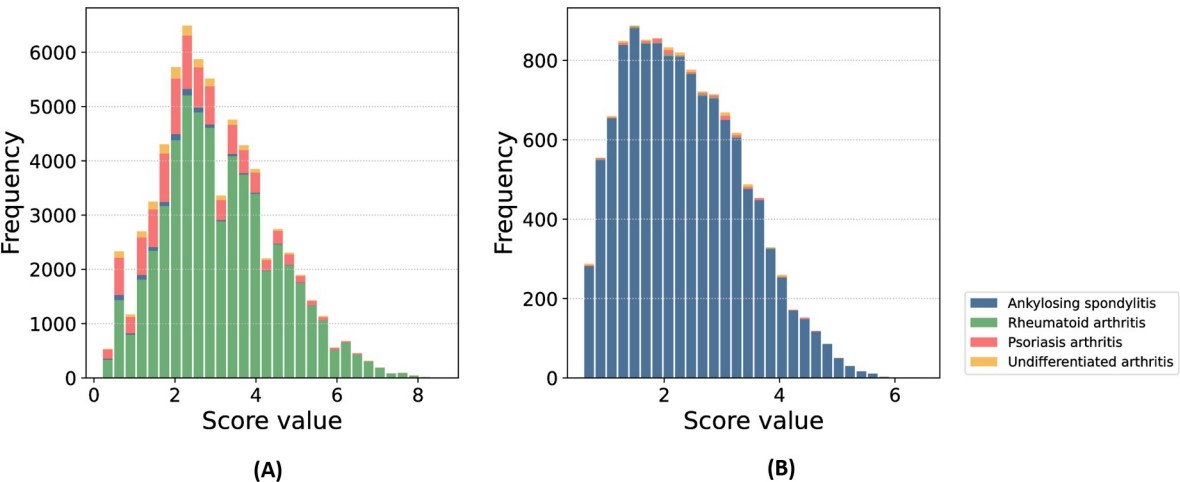

**Fig 1. Disease activity scores distribution.** Stacked histograms showing the DAS28 and ASDAS distribution in the preprocessed dataset. The different colour bars show the different arthritis types. **(A) DAS28 distribution.** The DAS28 score is usually recorded for patients with RA. **(B) ASDAS distribution.** The ASDAS score is usually recorded for patients with axSpA.

approach usually implies discarding most temporal information and using only the dataset's main features, leading to significant information loss, poor generalisability and bias.

With this in mind, our goal is to develop a deep learning model that can process the full patients' history, overcoming the challenges of temporal and feature irregularity. Moreover, it should be modular and support multiple outcome predictions allowing us to learn from all patients in the dataset with different DAS scores and arthritis subtypes. Lastly, it should produce meaningful latent representations, allowing us to compare patients with heterogeneous histories. An overview of the project pipeline, from data collection to implementation and evaluation of the different models is provided in Fig 2.

**2.2.2 Architecture.** Our model combines two main components. First, the model uses multilayer perceptrons (MLPs), long short-term memory networks (LSTMs) [6] and is augmented with attention layers [17] to build explainable vectorised patient representations. The different sources of information in the patient histories are handled separately until aggregation in the representation block. Then, we trained multilayer perceptrons to predict future DAS from these representations.

We adapted the architecture proposed in [15] to our setting by training multiple LSTMs, prediction networks, and by augmenting the model with several layers of attention layers. Fig 3 shows the model architecture with a brief description for each component of the model.

**Model input.** The input features are the patient medications, PROM and CMs up to a chosen time point, the demographics and the time to the prediction. Demographics, medications, PROM and CM are treated separately since their measurements are not aligned in time and contain different features. Merging them would result in a very sparse matrix and necessitate significant feature imputation.

**Model output.** The model predicts the next available DAS28 or ASDAS score by feeding the computed latent representation in the penultimate layers (i.e. representation layers) to two separate blocks of prediction layers. The latent representation is used posthoc to compute patient similarities.

**Encoders.** First, the MLP encoders process the normalised event-specific features. We defined separate encoders for each type of information (CM, Dem, PROM and Med). The

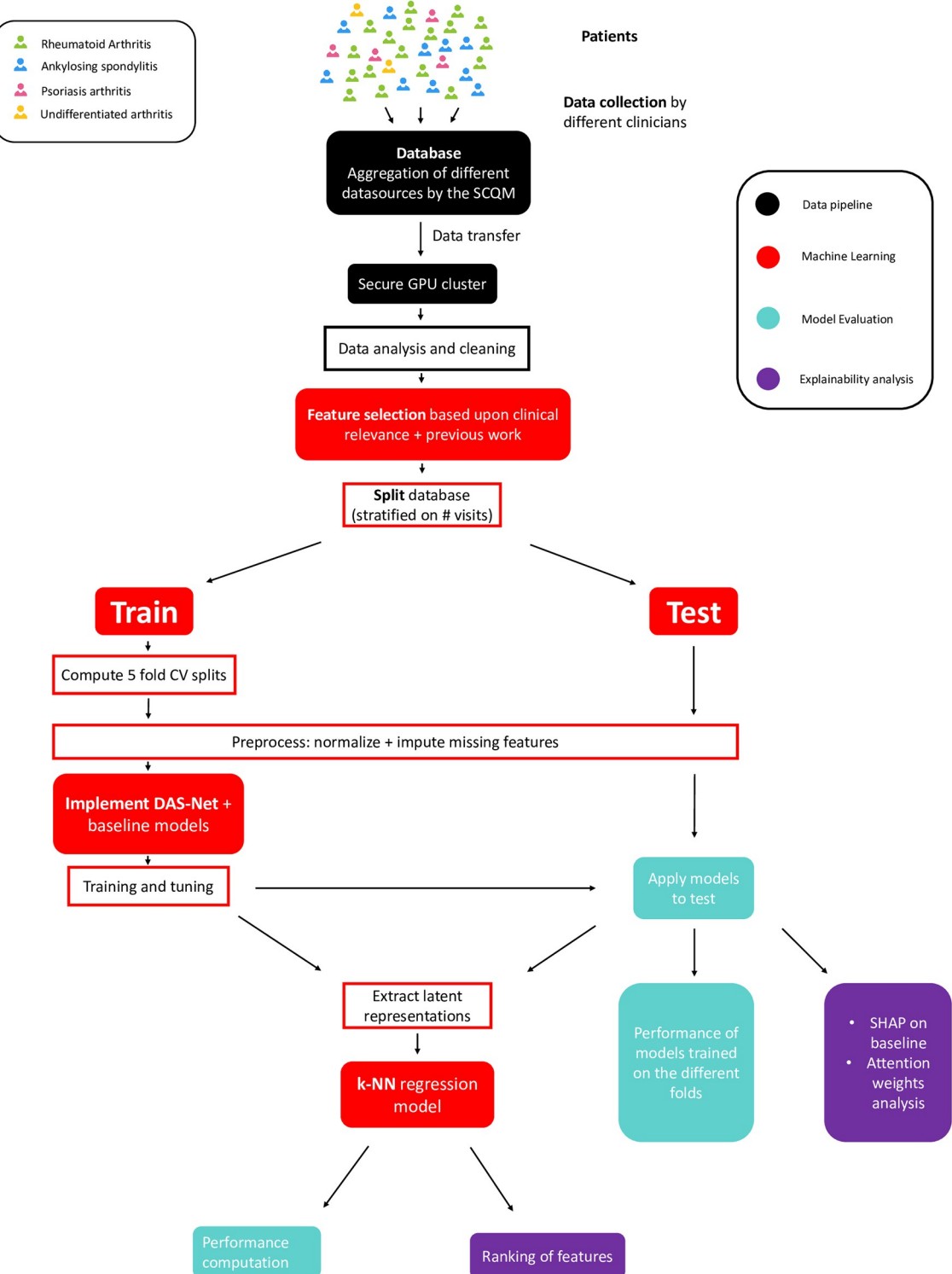

**Fig 2. Project pipeline from data collection to implementation and evaluation of the different models.**

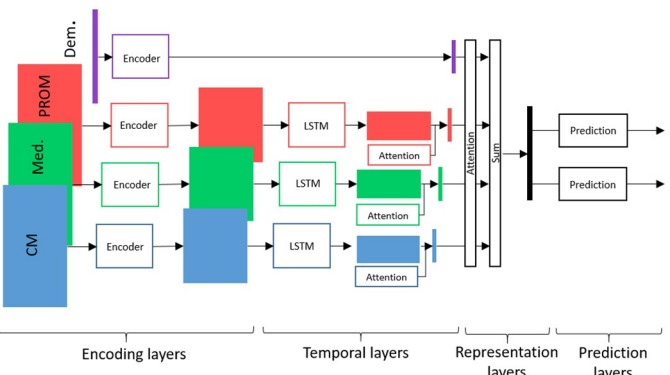

**Fig 3. Model architecture.** The encoders and prediction networks are MLPs. The model uses LSTMs to aggregate input sequences of different lengths and attention mechanism to weigh the different components of the input. "CM" stands for "Clinical measures", "Med." for "Medications", "PROM" for "Patient-reported outcome measure" and "Dem." for "Demographics".

encoders output lower dimensional embeddings for the time-related events and higher dimensional embeddings for the demographics to have matching history sizes in the later aggregation step. The order of the initial events is maintained in the computed embeddings.

We describe how the model is applied to a patient $p$. Let $ev \in \{CM, Med, PROM\}$ be a time-related event, $s_{ev}$ the number of features for $ev$, $q_{ev}$ the embedding size, $E_{ev} : \mathbb{R}^{s_{ev}} \to \mathbb{R}^{q_{ev}}$ be the corresponding encoder and $[X_{t_{1(ev)}}, \ldots, X_{t_{n(ev)}}]^T \in \mathbb{R}^{s_{ev} \times n(ev)}$ the ordered events measured at times $t_{1(ev)} < \ldots < t_{n(ev)}$. To ease the notation, we omitted the dependencies to $p$. We store the time-ordered embeddings $[e_{t_{1(ev)}}, \ldots, e_{t_{n(ev)}}]^T \in \mathbb{R}^{q_{ev} \times n(ev)}$ with $e_{t_{i(ev)}} = E_{ev}(X_{t_{i(ev)}})$.

For the demographics event, we simply have $e_{dem} = E_{dem}(X_{dem})$, where $X_{dem} \in \mathbb{R}^{s_{dem} \times 1}$ are the demographic features.

**Temporal block.** For a given sequence of events, the temporal block aggregates the embeddings into a one-dimensional vector. It contains one LSTM and one attention mechanism per category of time-related events. The LSTMs process the ordered embeddings computed by the event encoders. The attention mechanism is a trainable vector that weighs the contribution of each output of the LSTMs to the aggregated event history. For a given event, the aggregated history vector is the weighted sum of the outputs of the LSTM.

Thus, let $L_{ev}$ be the LSTM for event $ev$, $ev \in \{CM, Med, PROM\}$. $L_{ev}$ takes as input the sequence of embeddings $[e_{t_{1(ev)}}, \ldots, e_{t_{n(ev)}}]^T$ and outputs a processed sequence $[L_{ev}(e_{t_{1(ev)}}), \ldots, L_{ev}(e_{t_{n(ev)}})]^T$. Given the computed local attention weights $a_{t_{i(ev)}}^{loc}$, $i = 1, \ldots, n$, the aggregated event history is

$$H(ev) = \sum_{i=1}^{n} a_{t_{i(ev)}}^{loc} \cdot L_{ev}(e_{t_{i(ev)}}),$$

where using the *softmax* operator we have that $\sum_{i=1}^{n} a_{t_{i(ev)}}^{loc} = 1$.

**Representation block.** The representation block combines the event-specific outputs $H(ev)$ of the temporal block, the demographics embedding $e_{dem}$ and the time to prediction $t$ into a unique vector. It is augmented by an attention mechanism, weighing the contribution of each type of event to the representation. The representation of a patient is the weighted sum of the

demographics embedding and the aggregated event-specific histories, concatenated with the prediction time $t$.

Thus, $R = [P, t]$ where

$$P = \sum_{ev \in \{CM, Med, PROM\}} a^{glob}(ev) \cdot H(ev) + a^{glob}(dem) \cdot e_{dem}$$

$$\text{and} \sum_{ev \in \{CM, Med, PROM, Dem\}} a^{glob}(ev) = 1.$$

$R = [P, t]$ is the combined latent representation of the patient history. It is used as input to predict future disease states and to compute similarities between patients.

**Prediction networks.** We defined two multilayer perceptron prediction networks, $P_{DAS28}$ : $\mathbb{R}^{r \times 1} \to \mathbb{R}$ and $P_{ASDAS} : \mathbb{R}^{r \times 1} \to \mathbb{R}$. The networks take as input the patient representation $R$ and output the predicted DAS value at the medical visit at time $t$.

**2.2.3 Features and target selection.** As described in subsection 2.1.3, we only included patients with at least three measurements of either DAS28 or ASDAS. These two DAS are part of the clinical measures, i.e. they are recorded during the medical visits of the patients. We use as targets the DAS collected from the second CM onwards, to ensure sufficient history length. The DAS from past CMs are part of the input features; a DAS is thus the target and then a feature once it becomes part of the patient's history.

For each possible target, we used as input features the demographics and all the time-related events observed at least 15 days before the target CM.

**2.2.4 Optimisation.** We stratified the patients on the number of CMs and randomly sampled 20% of the stratified patients as testing set that was not used for model training and tuning. We standardised the features and imputed missing values. We performed a five-fold CV on the training data to find the optimal parameters via random search. We selected the hyperparameters with the lowest average loss across the folds on their respective validation sets.

Following the empirical risk minimisation principle, our training objective is the sum of the mean squared error (MSE) for the DAS28 and ASDAS predictions. We used the AdamW [18] algorithm with mini-batch processing to optimise the objective.

At each step, we randomly sampled two batches of patients, one containing the patients with available DAS28 and the other with available ASDAS to ensure consistent joint optimisation of both objectives for these patients. We predicted all the available targets for each selected patient. The loss optimised at each optimiser step is defined in Eq 1

$$L(\theta) = \sum_{B \in \{B_{DAS28}, B_{ASDAS}\}} \frac{1}{N_B} \sum_{p \in B} \sum_{v=1}^{n_p} \left( model_\theta(f_p^v, t_v) - y_p^v \right)^2$$

where $B_{DAS28}$ and $B_{ASDAS}$ are the sampled batches patients with available DAS28 and ASDAS respectively, $N_B$ is the total number of targets in batch $B$, $n_p$ is the number of targets for patient $p$, $f_p^v$ are the input features for patient $p$ to predict target $v$, $t_v$ is the time to target $v$ and $y_p^v$ is the true value of the target. $\theta$ denotes the model parameters to be optimised. We used batch sizes proportional to the total number of available targets per score to ensure consistent joint optimisation of both prediction networks.

## 2.3 Patient similarity: $k$–NN regression model

We evaluated the utility of DAS-Net's computed latent representations (i.e. computed vector representation $R$ as described in subsubsection 2.2.2) to retrieve similar patients. Given a patient representation at a prediction time-point, we computed the $L^1$ distance to all other

representations and selected the $k$ closest patient embeddings. We set $k$ to 50 as it achieved optimal performance on the validation data (appendix S5 Fig).

We matched the computed patient representations from the test set to their closest representations in the train set, such that for each patient representation $e_{p,t} \coloneqq e \in \mathcal{R}_{test}$ (i.e. the computed representation embedding for patient $p$ at time $t$), we found the subset of nearest neighbour representations $\mathcal{N}_e \in \mathcal{R}_{train}$. We omitted the dependencies to $p$ and $t$ to ease the notation. This experiment simulates comparing incoming data to an extensive established database, possibly across hospitals. It could help find optimal management strategies faster by assessing which strategy worked best for similar patients.

Analogous to $k$−NN regression, we compared the representation's future DAS with the average DAS of their closest matched set. We refer to this model as the $k$−NN regression model.

**2.3.1 Feature importance for similarity assessment.** We developed aggregate metrics to assess the average importance given to each feature for the similarity computation between an index patient and their subset of nearest neighbours.

For continuous features, we computed the average absolute distance (AAD) between the feature value of the patients in the test set and the average value in their matched set (in the training data), and the standardised AAD by dividing the AAD by the standard deviation of the feature:

$$AAD = \frac{1}{|\mathcal{R}_{test}|} \sum_{e \in \mathcal{R}_{test}} | \, x_e^c - \frac{1}{|\mathcal{N}_e|} \sum_{e' \in \mathcal{N}_e} x_{e'}^c \, |,$$

where $x_e^c$ is the value of the continuous feature $c$ for patient embedding $e$. For all computations, we restricted the subsets to the embeddings with available feature $c$. This metric reflects how much the values of the features of the subset of nearest neighbours deviate from the values of the index patient.

For a categorical feature $f_j$ with possible categories $S_j$ we computed the prior empirical probability of each category $k \in S_j$. Furthermore, for each $k \in S_j$, we computed the adjusted probabilities for the embeddings in the neighbourhood $\mathcal{N}_e$ of an index patient embedding $e$ with feature value $k$, i.e. the probability $P(x_{e'}^j = k \mid x_e^j = k, e' \in \mathcal{N}_e)$. We compared the two quantities to evaluate the importance of each categorical feature for the similarity computation. For an embedding $e' \in \mathcal{R}_{train}$, the prior empirical probability $P(x_{e'}^j = i)$ of category $i \in S_j$ is

$$P(x_{e'}^j = i) = \frac{\sum_{e \in \mathcal{R}_{train}} \mathbb{1}\{x_e^j = i\}}{\sum_{e \in \mathcal{R}_{train}} \sum_{k \in S_j} \mathbb{1}\{x_e^j = k\}},$$

and the adjusted probability is

$$P(x_{e'}^j = k \mid x_e^j = k, e' \in \mathcal{N}_e)) = \frac{\sum_{e \in \mathcal{R}_{test}} \mathbb{1}\{x_e^j = k\} \sum_{e' \in \mathcal{N}_e} \mathbb{1}\{x_{e'}^j = k\}}{\sum_{e \in \mathcal{R}_{test}} \mathbb{1}\{x_e^j = k\} \sum_{e' \in \mathcal{N}_e} \sum_{i \in S_j} \mathbb{1}\{x_{e'}^j = i\}}.$$

Again, we restricted the computations to the subsets of patients with available feature $j$. The increase in adjusted probabilities versus prior probabilities reflects how likely the feature is to have the same value as the index patient within its subset of nearest neighbours.

## 3 Results and discussion

We compared the performance of DAS-Net and of the $k$−NN regression model for future disease activity prediction to different baseline models and further explored the three

explainability approaches to better understand the relationship between input features and model output at different stages of the modeling process.

## 3.1 Performance

**3.1.1 DAS-Net prediction.** We compared the performance of our model to two non-temporal machine learning models: vanilla neural network (MLP) and tree-based gradient boosting model (XGBoost), and one temporal LSTM model. Furthermore, we also included a static naive baseline. The static naive baseline uses the last available DAS28 (resp. ASDAS) score for the given patient as its prediction. This strategy implies using the last disease state of a patient as a predictor of their future disease state. The MLP and XGBoost baselines take as input the same features as our model but only their last available values. Restricting the number of values per feature is necessary since these models cannot handle varying input sizes. We trained one MLP and XGBoost model per prediction task. Like our model, the LSTM baseline also uses the complete patient history as input. Besides the attention mechanism, the main difference between the LSTM and DAS-Net models lies in the disposition of the long short-term memory layers. DAS-Net employs separate long short-term memory layers for each type of event (CM, Med, PROM), while the LSTM model uses a unique long short-term memory layer to process the concatenated events.

In Table 1 we report the models' average performance and standard deviation on the test set. Our model achieves the lowest mean squared error (MSE) on both prediction tasks (MSEs of 0.510 ± 0.009 for ASDAS and 0.965 ± 0.014 for DAS28). In second place comes the LSTM model for ASDAS prediction (MSE of 0.521 ± 0.007) and the XGBoost model for DAS28 prediction (MSE of 0.992 ± 0.002 for DAS28). Using a naive model that uses the most recent DAS score as prediction achieves the worst performance (MSEs of 0.842 for ASDAS and 1.475 for DAS28).

Furthermore, we evaluated the models' ability to correctly predict active RA (i.e. DAS28 values above 2.6) and moderate axSpA (i.e. ASDAS values above 2.0). To perform the classification, we trained a logistic regression model on DAS Net's latent embeddings from the training set and evaluated the performance on the test set. We compared the performance of this approach to the LSTM, XGBoost and MLP predictions, where we thresholded the predicted values of DAS28/ASDAS. Our approach achieves overall a higher accuracy than the baseline ML models (accuracies of 0.761 ± 0.001 for ASDAS and 0.757 ± 0.000 for DAS28 for our approach) (Table 2). Furthermore, the sensitivity and specificity of our approach are more balanced than for the baseline models. The baseline models achieve a higher sensitivity but suffer from a low specificity (Table 2).

To understand the effect of the length of patient history on the prediction performance, we computed the model's performance as a function of varying lengths of patient histories. Fig 4 shows the MSE decreases as more prior medical visits become available to the model.

**Table 1. Model performance (regression).** DAS-Net outperforms the four baselines for both prediction tasks. The naive baseline simply reuses the last available DAS. The MLP and XGBoost baselines use the last available values of each feature as input and our model the whole patient history. The LSTM baseline sequentially processes the patients' histories.

| Model | MSE ASDAS | MSE DAS28 |
|---|---|---|
| DAS-Net | **0.510 ± 0.009** | **0.965 ± 0.014** |
| LSTM | 0.521 ± 0.007 | 1.011 ± 0.018 |
| XGBoost | 0.534 ± 0.003 | 0.992 ± 0.002 |
| MLP | 0.562 ± 0.005 | 1.029 ± 0.007 |
| Naive | 0.842 | 1.475 |

**Table 2. 3.1.2 Model performance (classification).** We evaluated the performance of the different approaches at predicting active disease (i.e. DAS28 values above 2.6 or ASDAS values above 2.0). While our approach has a slightly lower sensitivity than the baselines, it has a better balance between sensitivity and specificity and has an overall higher accuracy.

| Model | Sensitivity ASDAS | Specificity ASDAS | Accuracy ASDAS | Sensitivity DAS28 | Specificity DAS28 | Accuracy DAS28 |
|---|---|---|---|---|---|---|
| DAS-Net | 0.771±0.002 | **0.749 ± 0.003** | **0.761 ± 0.001** | 0.759±0.001 | **0.754 ± 0.001** | **0.757 ± 0.000** |
| LSTM | 0.800 ± 0.022 | 0.715 ± 0.025 | 0.759 ± 0.004 | 0.828 ± 0.027 | 0.639 ± 0.048 | 0.736 ± 0.010 |
| XGBoost | **0.859 ± 0.009** | 0.619 ± 0.014 | 0.750 ± 0.003 | **0.845 ± 0.001** | 0.620 ± 0.002 | 0.736 ± 0.001 |
| MLP | 0.818 ± 0.008 | 0.657 ± 0.010 | 0.745 ± 0.001 | 0.835 ± 0.009 | 0.631 ± 0.011 | 0.736 ± 0.002 |

Additionally, in Fig 5, we plot the predicted versus ground truth DAS28 and ASDAS scores for two example patients, showcasing how DAS Net could be used by clinicians to monitor and predict disease activity.

Lastly, in Appendix S1 Text we show that the model predictions are robust across subgroups of patients with different characteristics. We also present the results of additional experiments aiming to evaluate the impact of feature imputation on model predictions and to demonstrate the model's robustness to spurious correlations.

**3.1.2 Patient similarity: $k$−NN regression model.** We evaluated the ability of our model to cluster patients with similar disease progressions, by comparing the future DAS values of the embeddings in the test set with the average values of their most similar embeddings, as computed by our $k$−NN regression approach on DAS-Net's latent embeddings. We compared the performance of our approach to the performance of a $k$−NN algorithm applied to the raw data, and a naive approach selecting a random subset of patients (Table 3). Both baseline strategies thus do not utilise DAS-Net's computed latent representations. The $k$-NN model on the latent representations achieves the lowest MSE (MSEs of 0.506 and 0.966 for ASDSAS and DAS28 prediction).

Interestingly, our $k$−NN approach has a similar predictive performance to the DAS-Net model for prediction (Table 1), and also outperforms the LSTM, MLP and XGBoost baselines, suggesting that the DAS-Net latent representations successfully capture the important predictive components from the patient history.

## 3.2 Explainability approaches

In this section, we compare and contrast the results obtained from the different feature attribution techniques we applied or developed. These methods offer multiple insights on the

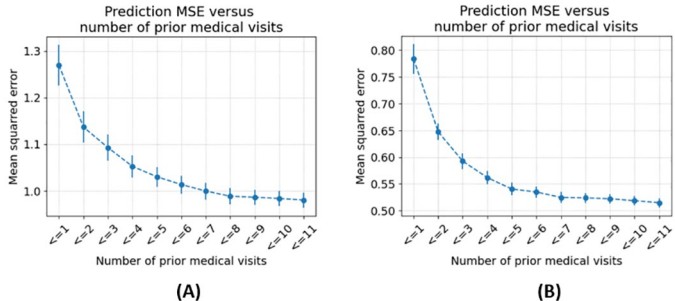

**(A)**                                              **(B)**

**Fig 4. MSE versus number of prior medical visits.** The MSE between model predictions and target DAS values decreases as the number of prior medical visits increases. The availability of at least three prior medical visits induces a steep decrease in MSE. Panel **(A)** shows the MSE for the DAS28 prediction and panel **(B)** for the ASDAS prediction.

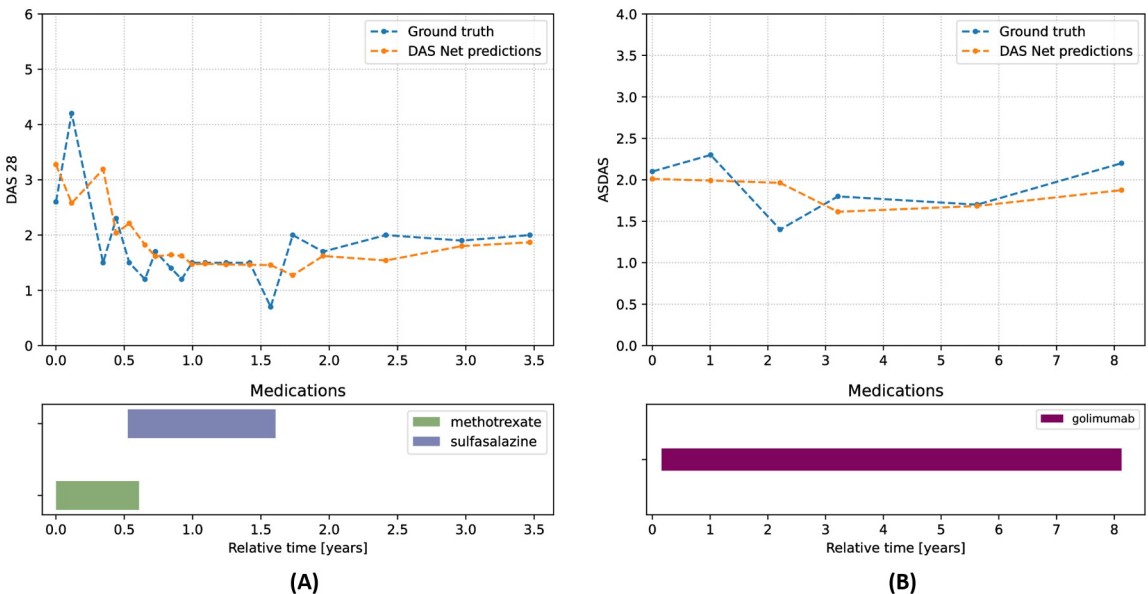

**Fig 5. Predictions of individual patient trajectories.** We compare the model predictions with the ground truth values of DAS28 (panel (**A**)) and ASDAS (panel (**B**)) for two example patients. The bar charts show the prescribed medications present in the database.

relationship between input features and model output at different stages of the modeling processes.

**3.2.1 SHAP values on vanilla neural network.**   For the baseline neural network model (MLP), we computed the SHAP [10] values for the input features. SHAP values are derived from the game-theoretic-based Shapley values [19] and compute the contribution of each feature to the model predictions.

The plots in Fig 6 show the top-10 SHAP values for ASDAS and DAS28 predictions. Each dot represents a feature value from the test set and is overlaid with a colour reflecting the value of the feature. The x-axis shows the SHAP value. In our setting, a positive SHAP value indicates that the feature drives the model predictions upwards, and thus leads to higher predicted DAS. The features are ordered by the average magnitude of their SHAP values (from top to bottom, and we included only the top ten features). Overall, the SHAP values are consistent with the clinical knowledge.

For ASDAS prediction, the *past ASDAS* values, *age* and *number of enthesitides* are positively correlated with their SHAP values, indicating that a higher value leads to a higher predicted disease activity score. For the medications, currently taking a *bDMARD* leads to lower future predicted DAS and the opposite for *csDMARDs*. For DAS28 prediction, the *past DAS28* values, *BSR*, *HAQ* and *RADAI pain level* are positively correlated with higher predicted disease activity

**Table 3. Similarity matching.** The *k*-NN (*k* = 50) method based on the model latent embeddings outperforms the *k*−NN algorithm directly applied to the raw data and the completely random subset for the retrieval of similar patients.

| Model | MSE ASDAS | MSE DAS28 |
|---|---|---|
| *k*-NN model on DAS-Net latent representations | **0.506** | **0.966** |
| *k*-NN on raw data | 0.681 | 1.218 |
| Random subset | 0.915 | 1.863 |

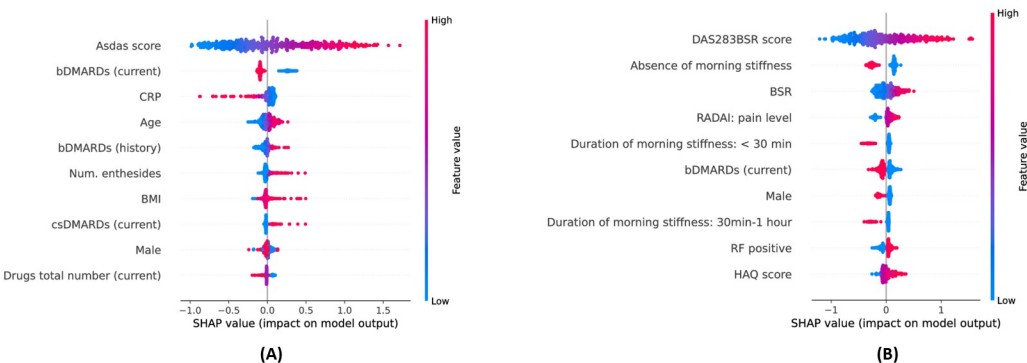

**Fig 6. SHAP feature importance.** The x-axis shows the SHAP value, and each dot is overlaid with a color representing the feature value. Thus, a pink dot with a positive SHAP indicates that the feature has a high value and leads to a higher predicted DAS. We show the top-10 features with the highest absolute SHAP values (ordered from top to bottom) for ASDAS prediction (panel (**A**)) and DAS28 prediction (panel **B**).

scores. The *absence or short duration of morning stiffness* leads to lower predicted DAS. Being *male* is also a better prognostic factor.

Furthermore, we computed the absolute SHAP values of the features for each model trained on one of the 5 folds in our data (during 5-fold cross-validation). The plots in Fig 7 show the average and standard deviation of the absolute SHAP values for the 10 features with the largest overall absolute SHAP values (ordered from top to bottom). The importance ranking of the features is consistent across the different models.

**Clinical relevance of findings**. In predicting future DAS in RA patients, the model was strongly influenced by the presence and duration of morning stiffness, with no or shorter morning stiffness resulting in lower predicted DAS. Morning stiffness for more than one hour strongly correlates with DAS28 scores [20]. Thus, in the model, the level of morning stiffness might have reinforced the strong dependency of the future DAS from current and past DAS measurements.

Notably, the feature importance in predicting ASDAS in patients with axSpA differed with respect to the influence of current and past treatment. In RA, current use of bDMARDs predicted low DAS levels. Similarly, in axSpA, the current use of bDMARDs was linked to predicting low future disease activity. This suggests that bDMARDs are effective in managing disease progression in this context. However, in the axSpA cohort, the situation is more complex. Both past use of bDMARDs and current use of csDMARDs (conventional synthetic disease-

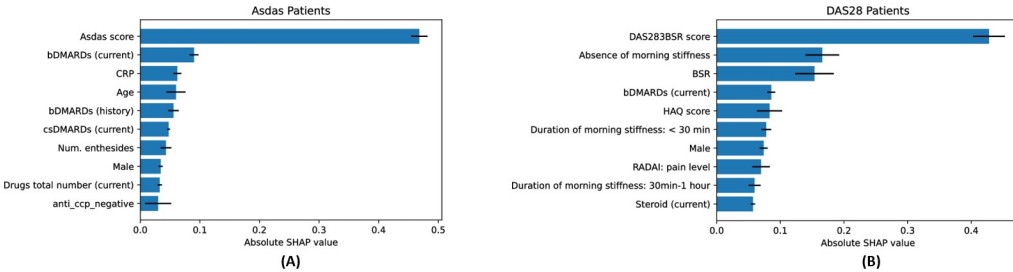

**Fig 7. Mean and standard deviation of the absolute SHAP values across folds.** We trained one MLP model per DAS on each cross-validation fold and computed their average absolute SHAP values on the test set for ASDAS prediction (panel (**A**)) and DAS28 prediction (panel **B**). The top-10 most important features are consistent across folds.

modifying antirheumatic drugs) are connected to high future disease activity. This suggests that patients who have experienced previous failure with bDMARDs or require additional csDMARD therapy belong to a difficult-to-treat group with a low likelihood of responding favourably to future treatments.

**3.2.2 Attention weights.** DAS-Net employs a two-layered attention mechanism for model-based explainability. The attention mechanism assign weights to the different events of the patient histories highlighting their significance for the model's predictions. The **local attention** is specific to each type of time-related event showing the weight given to each event when building the aggregated event history ($H(ev)$, $ev \in$ {*CM*, *Med*, *PROM*} in subsection 2.2.2). For example, they show which specific clinical measure contributed the most to the prediction. The **global attention** gives weight to the aggregated event histories and demographics when building the patient's full history representation ($P$ in subsection 2.2.2). It shows which type of event is used the most by the model to make the prediction.

**Global attention.** Fig 8A shows the attribution of the global attention weights to the different event features (i.e. CM, PROM, etc.) in the patients' history as the history length increases (denoted by the number of predicted targets). At the first target prediction, while most of the attention weight is already attributed to past CM, one-third is still attributed to other sources of information. Thus, when limited information is available, the model considers all the sources of information (i.e. clinical measures, medications, demographics and PROM). As the volume of available information increases (i.e. increasing length of history), the model increasingly assigns higher weights to the past clinical measures (CM) compared to the other sources of information. This weight distribution is reasonable because the previous CM contain the previous DAS that is predictive of future DAS.

Interestingly, for patients with a significant improvement in DAS (at least 20% improvement since the last CM), DAS-Net attributes less attention to the CM and redistributes it towards the other types of events (Fig 9).

**Local attention.** We further inspected the attribution of the local attention weights for the clinical measures in patients' history when predicting the target outcome Fig 8B. Most attention is directed at the last available clinical measure in the history before the prediction. Furthermore, the attribution to past clinical measures is inversely proportional to their distance from the target. Our model thus assigns the highest attention scores to the recent clinical measures (i.e. latest measures), particularly the ones preceding the prediction.

**3.2.3 Patient similarity. Case-based visualisations.** We visualised the patient representations by computing and plotting their two-dimensional t-SNE embeddings [21]. We plotted

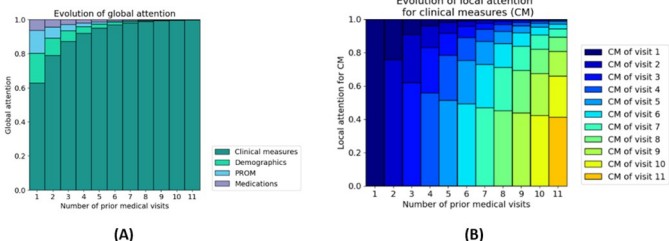

**Fig 8. Global and local attention weights for increasing number of medical visits (i.e. increasing patient histories) aggregated over the patients in the test set. (A) Global attention** weights for the different event features in the history. The global attention shows that the model uses clinical measures the most for the predictions. Furthermore, this pattern grows stronger as the number of available clinical measures increases. **(B) Local attention** weights for clinical measures. The local attention shows that within the clinical measures, most of the weight is attributed to the recent clinical measures.

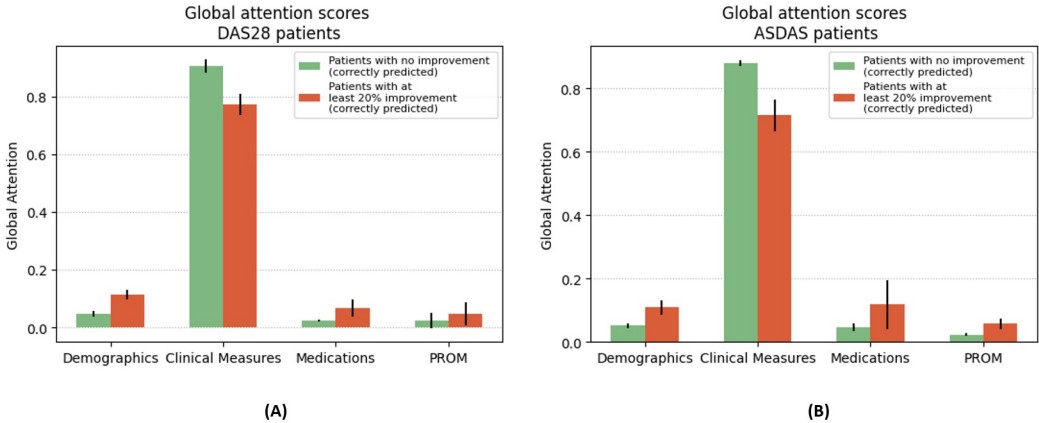

**Fig 9. Global attention weights.** Comparison in global attention weight attribution between patients with or without improvement in disease activity for DAS28 (panel (**A**)) and ASDAS (panel (**B**)). The attention is redistributed for patients with at least 20% improvement at the next visit.

the embeddings for the entire cohort, i.e. the t-SNE embeddings of all the higher dimensional representations in $\mathcal{R} = \mathcal{R}_{test} \cup \mathcal{R}_{train}$. In Fig 10, we overlaid the embeddings in each subplot with colourmaps reflecting the values of the features. We reported the last available value for the given feature at the time of computation of an embedding (we restricted the plots to the embeddings with an available value for the feature). The subspace is separated according to different values of the features. In Fig 10A, we overlaid the embeddings with the CIJD subtype of the patients, even though this attribute is not explicitly used as an input feature in our model, to get an overview of the distribution of the different CIJD subtypes in the latent space.

The plots provide general visual insight into the latent representation space. For instance Fig 10D shows the repartition of the smoker statuses, and a cluster of smoker patients in the top left of the figure stands out. Embeddings in this subspace correspond to patients with a smoking status that seems determinant for their disease activity prediction. Non-smoking patients and former smokers for more than a year are generally mapped to the same subspace, showing that the algorithm treats them the same. Some smokers, with possibly other more determinant factors, are also mapped in the same subspaces as non-smokers. By inspecting the gender plot (Fig 10C), we notice that males are generally mapped towards the edges of the sub-clusters. The same regions generally correspond to lower DAS28 activity regions (Fig 10B).

Furthermore, in Fig 10 we highlighted a randomly selected patient embedding $e_{p,t}$ from the test set (larger dot) and its nearest neighbours (triangles) $\mathcal{N}_e$ as computed by our $k$−NN regression model. For each continuous feature (here the *DAS28 score*) we also computed the average value in the entire representation set $\mathcal{R}$ and within $\mathcal{N}_e$. For categorical features (here *gender*, *duration of morning stiffness*, *rheumatoid factor* and *smoker status*), we computed the incidence of each category in $\mathcal{R}$ and $\mathcal{N}_e$. By comparing the overall distribution of the feature value with its distribution within $\mathcal{N}_e$, we get insight into the importance given to the different features for the similarity assessment.

The example patient in Fig 10 is diagnosed with rheumatoid arthritis, and most of her nearest neighbours also belong to the same CIJD subtype (Fig 10A). She has a higher *DAS28 value* than average (4.4 versus mean cohort value of 3.1) and there is a distribution shift within her subset of nearest neighbours towards higher DAS28 values (average of 4.2 within her subset of nearest neighbours) (Fig 10B). Her *smoker status* (Fig 10D) and *gender* (Fig 10C) seem determinant for the similarity assessment, since all of her nearest neighbours are also smoking

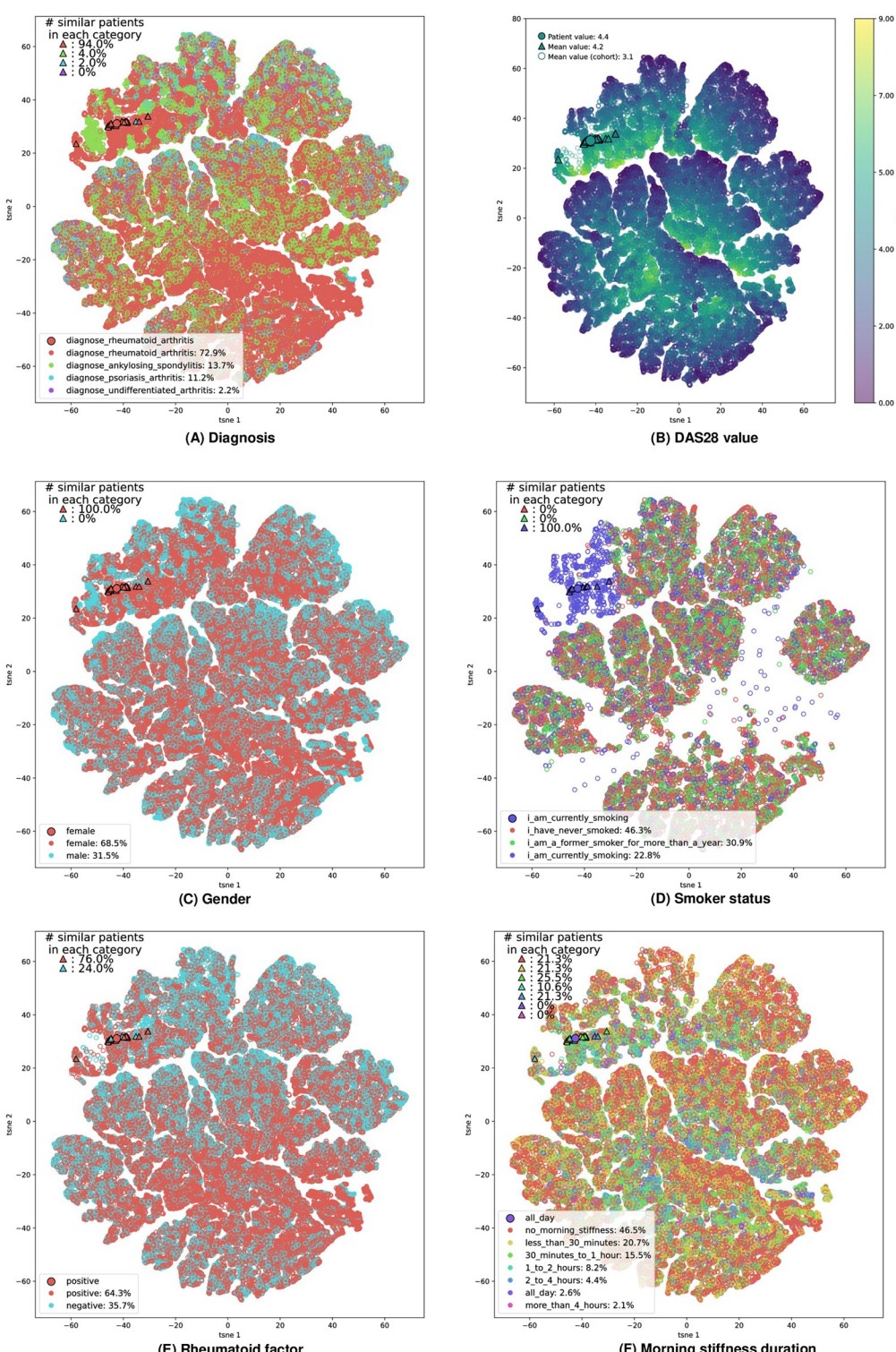

**Fig 10. t-SNE visualisation of patient representations.** Each point shows the t-SNE embedding of a representation of a patient at a given time. The subplots show the decomposition overlaid with the feature values (restricted to the embeddings with an available value for the feature). Furthermore, we highlighted a patient from the test set (larger filled dot) and her nearest neighbours (triangles) as computed by our algorithm. For each continuous feature we compute the average value in the entire cohort and within the subset of nearest neighbours. For categorical features, we computed the proportion of each category. We overlaid the plots with values representing different patient

characteristics; **(A)** Diagnosis, **(B)** DAS28 Value, **(C)** Gender, **(D)** Smoker status, **(E)** Rheumatoid factor, **(F)** Morning stiffness duration.

females. Conversely, the *rheumatoid factor* (positive, Fig 10E) and *duration of the morning stiffness* (all day, Fig 10F) seem to be considered less important for this patient. However, there is still an overall redistribution towards positive rheumatoid factor and longer durations of morning stiffness in the nearest neighbour subset compared to the distribution in the entire representation cohort. A similar analysis for different patients is provided in S6 and S7 Figs.

**Ranking of features.** Plots in Fig 10, S6 and S7 Figs provide insights into the nearest neighbour attribution mechanism on an individual patient level. Using the method described in subsection 2.3.1, we ranked the features by global importance in the cohort. We found that overall both DAS scores and the number of swollen joints are the most important for the similarity assessment for continuous features (Table 4). Similarly, high duration of morning stiffness and gender are the top-2 categorical features for the similarity assessment (Table 5).

**Clinical relevance of findings.** Our analysis of patient similarity suggested that the impact of smoking on disease parameters varies among patients. Genetic association studies showed that smoking is only associated with an increased risk of developing RA in people carrying the shared epitope genes in the HLA-DR locus, but not in current smokers without these RA risk genes [22]. While it is known that smoking negatively affects treatment response and disease severity in both RA and axSpA [23–26], it would be interesting to know if this is the same in all patients or if genetic background plays a similarly important role in the impact of smoking on disease.

**3.2.4 Use case.** In the previous sections, we demonstrated the different explainability layers that our analysis offers and highlighted the key cohort insights derived from them. Here, we present a final use case, showcasing the practical application of these different explanations for clinical decision-making. We revisit the prediction curve for the patient from Fig 5A,

**Table 4. Similarity metric: Contribution of continuous features.** Average absolute distance (AAD) and standardised AAD between the feature value of a test embedding $e_{p,t}$ and the mean feature value within its nearest neighbours $\mathcal{N}_e$. The features are ordered by standardised AAD. We see that the two DAS and the number of painful joints are taken into account the most during the similarity assessment.

| Feature | AAD | Standardised AAD |
|---|---|---|
| asdas_score | 0.25 | 0.24 |
| das283bsr_score | 0.35 | 0.25 |
| n_painfull_joints_28 | 2.05 | 0.41 |
| n_painfull_joints | 2.40 | 0.43 |
| crp | 5.35 | 0.46 |
| n_swollen_joints | 2.14 | 0.50 |
| bsr | 8.03 | 0.50 |
| mda_score | 0.73 | 0.56 |
| n_enthesides | 1.42 | 0.58 |
| joints_type | 8.05 | 0.61 |
| haq_score | 0.46 | 0.65 |
| pain_level_today_RADAI | 1.91 | 0.71 |
| activity_of_rheumatic_disease_today_RADAI | 1.89 | 0.71 |
| hb | 0.97 | 0.72 |
| height_cm | 6.73 | 0.73 |
| weight_kg | 12.05 | 0.76 |

**Table 5. Similarity metric: Contribution of categorical features.** Empirical probability of a category *c* versus adjusted probability, given that the data point is in the subset of nearest neighbours $\mathcal{N}_e$ of a datapoint $x_e$ with the same category *c*. The increase in the adjusted probability reflects the importance of a given category in the similarity assessment. Longer durations of morning stiffness and gender have the strongest impact on the similarity assessment.

| Category *c* | Base $P(c)$ | Adjusted $P(c|x_e = c)$ | Increase (percentage) |
|---|---|---|---|
| morning_stiffness_duration_RADAI: 2_to_4_hours | 0.04 | 0.08 | 100.0 |
| morning_stiffness_duration_RADAI: more_than_4_h... | 0.02 | 0.04 | 100.0 |
| gender: male | 0.29 | 0.46 | 59.0 |
| morning_stiffness_duration_RADAI: 1_to_2_hours | 0.08 | 0.11 | 38.0 |
| morning_stiffness_duration_RADAI: all_day | 0.03 | 0.04 | 33.0 |
| smoker: i_am_currently_smoking | 0.23 | 0.27 | 17.0 |
| ra_crit_rheumatoid_factor: negative | 0.37 | 0.43 | 16.0 |
| morning_stiffness_duration_RADAI: 30_minutes_to... | 0.16 | 0.18 | 12.0 |
| gender: female | 0.71 | 0.78 | 10.0 |
| morning_stiffness_duration_RADAI: no_morning_st... | 0.47 | 0.51 | 9.0 |
| ra_crit_rheumatoid_factor: positive | 0.63 | 0.68 | 8.0 |
| smoker: i_am_a_former_smoker_for_more_than_a_year | 0.31 | 0.33 | 6.0 |
| anti_ccp: negative | 0.38 | 0.40 | 5.0 |
| anti_ccp: positive | 0.62 | 0.63 | 2.0 |
| smoker: i_have_never_smoked | 0.46 | 0.47 | 2.0 |
| morning_stiffness_duration_RADAI: less_than_30_... | 0.21 | 0.21 | 0.0 |

hereafter referred to as "index patient", in Figs 11 and 12, while additionally showing the mean and standard deviation predicted by the $k-$ nearest neighbours. The feature importance derived from the patient similarity provides insights on model predictions at different time points. We analyse two specific prediction time points, highlighted by red rectangles in the trajectory plots in Figs 11A and 12A (representing the $7-$th and $16-$th predictions respectively), and examine the feature values within the subset of most similar patients at these times.

First, we visualise the latent trajectories of the patient and their nearest neighbors at two distinct prediction time points, as depicted in Figs 11B and 12B. Moreover, the heatmaps in Figs 11C, 11D, 12C and 12D compare the index patient's feature values to those of the nearest neighbours. For continuous features, we computed the average values within the set of nearest neighbours and for categorical features we computed the proportion of patients sharing the same category as the index patient. Rows of similar colors in the heatmaps indicate consistency in feature values between the index patient and their nearest neighbours. Thus, from Fig 11 we can deduce that the prediction at this time point was mainly driven by the patients' gender, MDA/DAS28 scores, and the affected joints. Similarly, the $16-th$ prediction illustrated in Fig 12 was mainly influenced by the RADAI duration of morning stiffness and the MDA score. Together, these different analyses and visualisations collectively enhance our understanding of the data driving the model predictions.

## Conclusion

In this work, we propose DAS-Net, a multitask neural network-based model for transforming heterogeneous rheumatic disease registry data into comparable patient representations and predicting future disease activity. When predicting future DAS, DAS-Net outperformed all non-temporal baseline models that discarded or simplified most of the patient history. Furthermore, it also outperformed a temporal LSTM model suggesting that DAS-Net is better suited to handle heterogeneous temporal patient records.

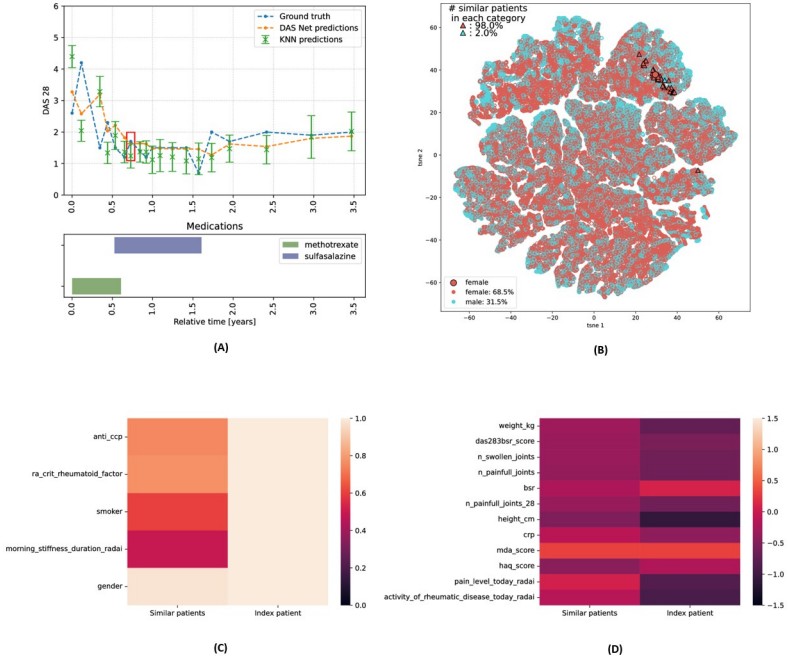

**Fig 11. Use case: Prediction at 7-th time point. (A) DAS28 predictions** of DAS-Net and $k$−NN models. **(B) t-SNE** overlaid with gender. **(C) Categorical features:** proportion of similar patients with same category value. **(D) Continuous features:** average values for similar patients.

Our model design included attention layers that aided in explaining the importance of the different visits and parts of the patient's history in outcome prediction. It showed that our model uses recent information but still attributes significant weight to older events and that the model attributes the majority of the weight to the clinical measures. This pattern gets stronger as the amount of available history increases and the model performance improves for longer medical histories.

Moreover, the predictive power of the nearest neighbour approach on the model's latent representations showed that our model is well suited to transform heterogeneous electronic health records into comparable representations. One possible extension for our model would be to explicitly incorporate a clustering loss in the training objective [27] to further improve the patient similarity framework.

Lastly, the results of the three different analyses of feature importance (feature attribution via SHAP, attention weights and case-based similarity) are in concordance with clinical expert knowledge ([28–30]). Past disease activity scores were consistently the strongest predictors in all three analyses and gender and rheumatoid factor stood out as important features for the similarity assessment. Consistent with these findings, low disease activity, including low CRP/BSR levels, and low HAQ levels have also been associated with good future outcomes in patients with RA in previous studies [31, 32]. Similarly, autoantibody status and gender have been described before as predictors of outcomes in RA patients [32–34]. Importantly, this analysis could be expanded to evaluate the influence of additional features not currently in the database, which might be linked to disease activity, such as ethnicity, and their effects on model predictions [35].

Overall, our study demonstrates promising results towards developing an explainable clinical decision support system for retrieving similar patients and predicting their disease

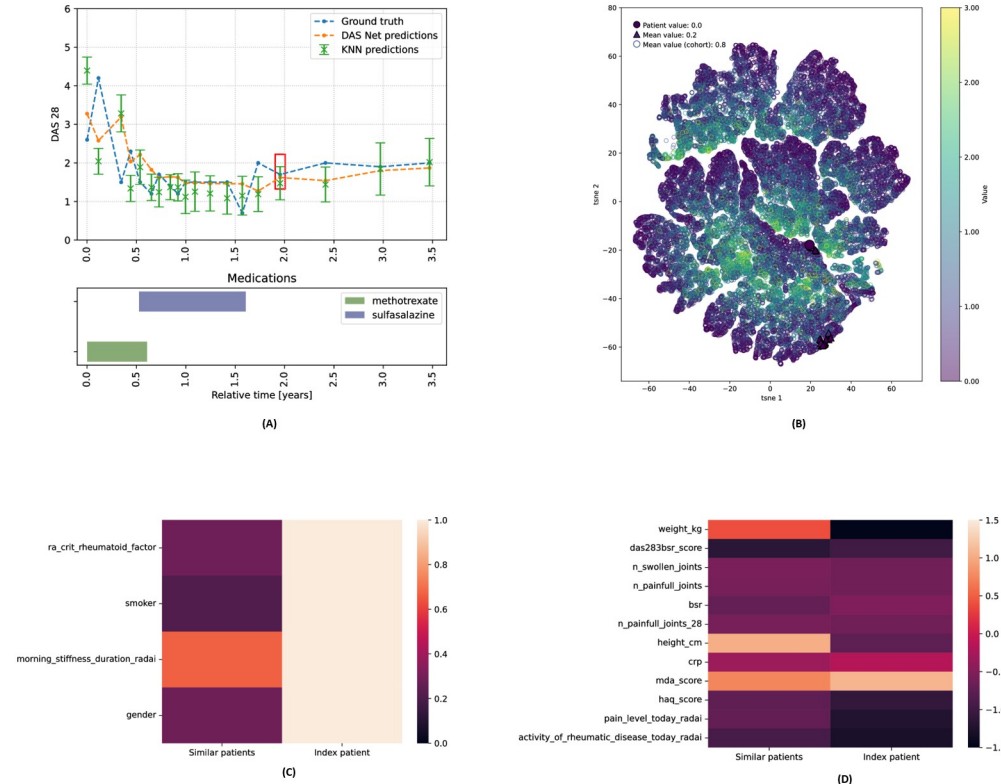

**Fig 12. Use case: Prediction at 16-th time point. (A) DAS28 predictions** of DAS-Net and $k$−NN. **(B) t-SNE** overlaid with HAQ score. **(C) Categorical features:** proportion of similar patients with same category value. **(D) Continuous features:** average values for similar patients.

progression while considering the different disease management strategies that worked best for similar patients. Such a CDSS would be especially useful for managing complex chronic diseases. It could help find optimal management strategies faster by assessing which strategy worked best for similar patients.

## Supporting information

**S1 Table. Name of input features in the different types of patient records in the SCQM database.**
(PDF)

**S2 Table. Mean, standard deviation and % missingness of continuous clinical measure features.**
(PDF)

**S3 Table. Distributions of categorical clinical measure features.**
(PDF)

**S4 Table. Mean, standard deviation and missingness of continuous medication features.**
(PDF)

**S5 Table. Distribution of categorical medication features.**
(PDF)

**S6 Table. Mean, std and missingness of continuous PROM features.**
(PDF)

**S7 Table. Distribution of categorical PROM features.**
(PDF)

**S8 Table. Mean, standard deviation and missingness of continuous demographic features.**
(PDF)

**S9 Table. Distribution of categorical demographic features.**
(PDF)

**S1 Fig. Distribution of the number of visits per patient.**
(PDF)

**S2 Fig. Feature distributions stratified on number of visits.**
(PDF)

**S3 Fig. Feature distributions stratified on number of visits (continued).**
(PDF)

**S4 Fig. Average scores versus number of medical visits.**
(PDF)

**S5 Fig. MSE versus $k$ in $k$−NN on validation data. We set $k$ to 50 in our final model.**
(PDF)

**S6 Fig. Patient similarity.** The DAS28 score and gender show a high level of consistency among the closest neighbors of this patient. Duration of morning stiffness and rheumatoid factors also show slight distribution shifts within the subsets of nearest neighbours.
(PDF)

**S7 Fig. Patient similarity.** The patient's rheumatoid factor status and their low DAS28 value can be observed among the nearest neighbours subset as well.
(PDF)

**S1 Text. Impact of features on prediction.**
(PDF)

## Acknowledgments

The authors thank the patients and caregivers who made the study possible, as well as the clinicians who collected the data. A list of rheumatology offices and hospitals that are contributing to the SCQM registries can be found on www.scqm.ch/institutions. The authors thank Almut Scherer for her feedback on the manuscript.

## Author Contributions

**Conceptualization:** Michael Krauthammer, Caroline Ospelt.

**Data curation:** Cécile Trottet, Aron N. Horvath.

**Formal analysis:** Cécile Trottet.

**Funding acquisition:** Michael Krauthammer.

**Investigation:** Cécile Trottet, Ahmed Allam, Raphael Micheroli, Michael Krauthammer, Caroline Ospelt.

**Methodology:** Cécile Trottet, Ahmed Allam.

**Project administration:** Raphael Micheroli, Michael Krauthammer, Caroline Ospelt.

**Resources:** Michael Krauthammer.

**Software:** Cécile Trottet, Aron N. Horvath.

**Supervision:** Ahmed Allam, Raphael Micheroli, Michael Krauthammer, Caroline Ospelt.

**Validation:** Cécile Trottet.

**Visualization:** Cécile Trottet.

**Writing – original draft:** Cécile Trottet, Ahmed Allam, Raphael Micheroli, Michael Krauthammer, Caroline Ospelt.

**Writing – review & editing:** Cécile Trottet, Ahmed Allam, Axel Finckh, Thomas Hügle, Sabine Adler, Diego Kyburz, Raphael Micheroli, Michael Krauthammer, Caroline Ospelt.

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
