## [Decision Letter · Decision Letter 0]

8 Apr 2024

PDIG-D-23-00447

Explainable deep learning for disease activity prediction in chronic inflammatory joint diseases

PLOS Digital Health

Dear Dr. Trottet,

Thank you for submitting your manuscript to PLOS Digital Health. After careful consideration, we feel that it has merit but does not fully meet PLOS Digital Health's publication criteria as it currently stands. Therefore, we invite you to submit a revised version of the manuscript that addresses the points raised during the review process.

Please submit your revised manuscript within 60 days Jun 07 2024 11:59PM. If you will need more time than this to complete your revisions, please reply to this message or contact the journal office at digitalhealth@plos.org. Please include the following items when submitting your revised manuscript:

We look forward to receiving your revised manuscript.

Kind regards,

Wisit Cheungpasitporn, MD

Academic Editor

PLOS Digital Health

Journal Requirements:

1. We ask that a manuscript source file is provided at Revision. Please upload your manuscript file as a .doc, .docx, .rtf or .tex.

2. Please provide separate figure files in .tif or .eps format only and remove any figures embedded in your manuscript file. Please also ensure that all files are under our size limit of 10MB.

Additional Editor Comments (if provided):

The authors of the study introduce DAS-Net, an innovative explainable deep learning model designed to forecast future disease activity in patients with chronic inflammatory joint diseases (CIJDs), utilizing a dataset from the Swiss Clinical Quality Management (SCQM) registry comprising longitudinal information for over 19,000 patients. DAS-Net integrates neural networks with attention mechanisms to analyze the heterogeneous medical histories of patients, aiming to predict future scores on the DAS28 and ASDAS metrics. Emphasizing explainability, the model employs multiple techniques to illuminate the decision-making process, highlighting its superior performance over non-temporal baseline models in predicting disease activity and its capability to identify latent representations for patients with similar disease progression patterns. Despite its strengths, concerns were raised regarding the model's potential biases, the representativeness of the SCQM dataset, and the generalizability of findings due to reliance on a single registry, suggesting a need for validation across diverse patient cohorts and incorporation of a broader range of patient data to enhance robustness and applicability in real-world settings.

Reviewers commend the research's ambition, clarity in presentation, and emphasis on model explainability, noting its potential to influence future patient care and aid clinical interpretation. However, they recommend several revisions to address gaps in the study. These include a more detailed description of the dataset, particularly in relation to ethnicity and patient visits, questioning the choice of a random data split, suggesting more rigorous comparator models, and advocating for clearer differentiation of the model's clinical significance. Further, improvements in the methodology for feature selection, a deeper examination of patient similarity metrics, and enhanced transparency in model predictions were advised. To address biases and enhance the model's clinical utility, incorporating domain expertise for causality assessment, expanding the variety of input features, and conducting prospective testing for real-world efficacy were among the key suggestions, aiming to bolster the model's relevance and trustworthiness in clinical settings.

Reviewers' comments:

Reviewer's Responses to Questions

**Comments to the Author**

1. Does this manuscript meet PLOS Digital Health’s publication criteria? Is the manuscript technically sound, and do the data support the conclusions? The manuscript must describe methodologically and ethically rigorous research with conclusions that are appropriately drawn based on the data presented.

Reviewer #1: Yes

Reviewer #2: Yes

2. Has the statistical analysis been performed appropriately and rigorously?

Reviewer #1: Yes

Reviewer #2: Yes

3. Have the authors made all data underlying the findings in their manuscript fully available (please refer to the Data Availability Statement at the start of the manuscript PDF file)?

Reviewer #1: Yes

Reviewer #2: No

4. Is the manuscript presented in an intelligible fashion and written in standard English?

Reviewer #1: Yes

Reviewer #2: Yes

5. Review Comments to the Author

Reviewer #1: The authors propose an explainable deep learning model called DAS-Net to predict future disease activity in patients with CIJDs. CIJDs cause inflammation and damage to joints over time. The model was trained on a dataset from the SCQM registry containing longitudinal data for over 19,000 patients. DAS-Net combines neural networks and attention mechanisms to process patients' heterogeneous medical histories and predict future scores on the DAS28 and ASDAS. Multiple explainability techniques were used to understand how the model makes predictions. The model outperformed non-temporal baselines in predicting disease activity. It also effectively extracted meaningful latent representations to identify patients with similar disease progression patterns. The various explainability analyses provided insights into the relationship between patients' characteristics, management history, and predicted disease outcomes. The findings overall demonstrate the promise of explainable deep learning methods to gain new evidence and support clinical decision-making for complex chronic diseases like CIJDs through predictive modeling and analysis of real-world patient data.

The dataset comes from a single registry in Switzerland. The patient population within this registry may have unique characteristics that do not reflect CIJD patients in general. For example, treatment approaches may differ. Before relying on the model predictions clinically, the model should be validated on entirely different patient cohorts from other registries or healthcare systems.

The SCQM dataset likely contains various biases, confounders, and influencing factors that impact disease progression markers and treatment decisions in ways the model does not account for. For instance, certain demographics like patient age may naturally correlate with higher disease activity. The model may make predictions based on these correlations without fully disentangling the true underlying relationships. Techniques like adding negative controls or causality assessment using domain expertise would help mitigate issues caused by biases and confounders in the real-world data.

By only selecting patients with certain scores like DAS28 and ASDAS recorded, a significant subset of patients are excluded which reduces the robustness and generalizability of findings. Ideally, a disease progression model would work reliably for any patient regardless of which particular scores they happened to have recorded over time. The model could be enhanced to extract signal from all patient journeys rather than just those with commonly recorded scores.

While reasonable feature selection was done, a more thorough analysis may reveal better input features for predicting disease activity and patient similarity assessments. Advanced feature learning methods could help extract non-linear representations and temporal abstractions instead of using raw measurements directly. Feature engineering tailored specifically to the predictive tasks could likely improve performance.

Evaluations focused heavily on overall performance metrics like accuracy for disease activity level prediction. However, model adoption ultimately depends on clinical utility in terms of patient health impact and changes to clinician decision making. Clinical trial-like analysis is needed to specifically demonstrate the benefits and harms of using the model predictions to guide treatment plans versus standard-of-care.

While attention weights provide basic explanations about which visits and measurements influence the model, translating those low-level attention values into intuitive explanations that directly resonate with physicians remains challenging. Natural language or visual interpretation methods tailored to connect attention mechanisms with clinical significance could make the explanations much more useful in practice.

The similarity analysis is primarily mathematical, based on comparing clinical measurements. Incorporating biological knowledge like inflammatory markers or genetic information into the similarity metric would likely improve the pairing of patients for comparison. This could extract disease subtypes more reliably than mathematical groupings alone.

Several explainability approaches were used but not rigorously quantified in terms of their impact on model transparency or physician understanding. Surveys, questionnaires, and other metrics are needed to systematically evaluate qualities like trust, mental model alignment, and decision impact for the explanations. This would better differentiate better explanation methods.

The case-based explainability analyses were limited to a couple example patients. To better evaluate model behaviors, analyzing larger, more diverse patient subsets would provide greater insights into how the model handles nuances. Possible problematic explanations could also be identified through wider analysis.

Prospective testing within clinical environments compared to physician judgment would provide the highest-quality evidence of model efficacy and safety before full deployment. Additionally, model robustness checks and ongoing performance monitoring would be crucial to ensuring consistent predictive performance in varied situations of real-world use after deployment.

11. One major area of concern is potential biases in the model. The model is trained on observational data from a single registry. There are likely many confounders and selection biases affecting which patients get recorded, what data is measured about them, and treatments they receive over time. For example, age naturally correlates with higher disease activity. The model may end up making predictions based primarily on these data biases rather than true biochemical and inflammatory mechanisms. As the engineer, I would perform causal analysis to quantify the types and degree of biases, and use techniques like negative controls to minimize their impact on model decisions.

12. Additionally, the exclusion of patients without certain scoring measurements is problematic, as we want a model that works reliably for any patient. The loss of data further limits the model’s real-world usefulness. I would enhance the model to extract signals from all patient journeys, regardless of which particular scores are recorded. This is an engineering challenge but will reduce biases.

13. There are some transparency concerns around how the model arrives at predictions and similarities between patients. While attention weights provide some visibility into the model, it is hard to interpret those low-level signals. As the engineer responsible for deploying to clinicians, I would prioritize developing intuitive explanations from attention weights that directly connect to clinical variables and known disease mechanisms. This is crucial for establishing trust.

14. From a methodology perspective, the model evaluation is narrowly focused on overall predictive performance rather than clinical utility. Prospective testing is needed to prove the model actually improves outcomes and changes decision making before deployment. Additionally, ongoing monitoring of performance across diverse patient subgroups would be essential to making this a robust clinical system.

Reviewer #2: This paper describes the development of DAS-Net, a multi-task ML model to predict two established disease activity scores (DAS28 and ASDAS) in CIJD. Unique aspects to their approach include that the model incorporates various subtypes of CIJD (e.g. RA and axSPA) and through the use of attention layers the contribution of various features (e.g. demo, patient reported outcomes like morning stiffness, medications, etc.). 

There are several things worth commending: first, the aim of the research is of general interest as such ML strategies could in the future inform patient care or assist doctor interpretation of patients; second, the paper is very well and clearly written which is often challenging especially in this field; third, the author emphasize explainability or interpretability of their model and through the patient similarity provide a way to contextualize a patient amongst many; and fourth, the use a unique and large patient database of ~20k patients.

There are however several revisions I would recommend:

1. the dataset should be described more. There is an association between ethnicity and CJID, yet no ethnicity is provided in demographics (suppl Tabl 1). If that is not reported, it should at least be discussed in the conclusion. Further, a barchart showing the number of patients by number of visits is important. As the results show Figure 4, prediction improve markedly for patients with 3+ visits. Also, a breakdown of DAS28 and ASDAS and the other inputs (CM, PROM, DEMO, MEDs) for subjects with 3+ vs. up to 3 visits would be useful as I would like to know if there is a skew between these populations. 

2. I do not understand why the data split was random. Would it not be better t

---

## [Decision Letter · Decision Letter 1]

27 May 2024

Explainable deep learning for disease activity prediction in chronic inflammatory joint diseases

PDIG-D-23-00447R1

Dear Mrs Trottet,

We are pleased to inform you that your manuscript 'Explainable deep learning for disease activity prediction in chronic inflammatory joint diseases' has been provisionally accepted for publication in PLOS Digital Health.

Best regards,

Wisit Cheungpasitporn, MD

Academic Editor

PLOS Digital Health

I reviewed the revised manuscript and responses to reviewers comments. It appears that all comments have been appropriately responded to. I have no further comments and recommend publication.

Reviewer Comments (if any, and for reference):

Reviewer's Responses to Questions

**Comments to the Author**

1. If the authors have adequately addressed your comments raised in a previous round of review and you feel that this manuscript is now acceptable for publication, you may indicate that here to bypass the “Comments to the Author” section, enter your conflict of interest statement in the “Confidential to Editor” section, and submit your "Accept" recommendation.

Reviewer #1: All comments have been addressed

2. Does this manuscript meet PLOS Digital Health’s publication criteria? Is the manuscript technically sound, and do the data support the conclusions? The manuscript must describe methodologically and ethically rigorous research with conclusions that are appropriately drawn based on the data presented.

Reviewer #1: Yes

3. Has the statistical analysis been performed appropriately and rigorously?

Reviewer #1: Yes

4. Have the authors made all data underlying the findings in their manuscript fully available (please refer to the Data Availability Statement at the start of the manuscript PDF file)?

Reviewer #1: Yes

5. Is the manuscript presented in an intelligible fashion and written in standard English?

Reviewer #1: Yes

6. Review Comments to the Author

Reviewer #1: The revisions made by the authors comprehensively address the main points from the initial review. The expanded experiments, error analyses, and use case demonstration strengthen the rigor and clinical pertinence of the work. I believe the revised manuscript is of publishable quality, with the new content enhancing the overall impact and validity of the work.

7. PLOS authors have the option to publish the peer review history of their article (what does this mean?). If published, this will include your full peer review and any attached files.

**Do you want your identity to be public for this peer review?** For information about this choice, including consent withdrawal, please see our Privacy Policy.

Reviewer #1: No
